

# Generative learning for the problem of critical slowing down in lattice Gross-Neveu model

**Ankur Singha[1], Dipankar Chakrabarti[1] and Vipul Arora[2]**

**1** Department of Physics, Indian Institute of Technology Kanpur, Kanpur-208016, India
**2** Department of Electrical Engineering, Indian Institute of Technology Kanpur, Kanpur-208016, India

## Abstract

In lattice field theory, Monte Carlo simulation algorithms get highly affected by critical slowing down in the critical region, where autocorrelation time increases rapidly. Hence the cost of generation of lattice configurations near the critical region increases sharply. In this paper, we use a Conditional Generative Adversarial Network (C-GAN) for sampling lattice configurations. We train the C-GAN on the dataset consisting of Hybrid Monte Carlo (HMC) samples in regions away from the critical region, i.e., in the regions where the HMC simulation cost is not so high. Then we use the trained C-GAN model to generate independent samples in the critical region. We perform both interpolation and extrapolation to the critical region. Thus, the overall computational cost is reduced. We test our approach for Gross-Neveu model in 1+1 dimension. We find that the observable distributions obtained from the proposed C-GAN model match with those obtained from HMC simulations, while circumventing the problem of critical slowing down.

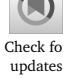

# 1   Introduction

Lattice field theory is the most reliable and well established technique to solve quantum field theories nonperturbatively. In this approach the theory is formulated on a discrete space-time lattice to solve numerically. In Monte Carlo(MC) simulation of lattice field theory the efficiency of simulation depends on the algorithm used. Algorithm like Hybrid Monte Carlo (HMC) [1] works well away from the critical points of the lattice theory but when one approaches the critical region the simulation algorithm suffers severe critical slowing down [2,3]. Near the critical point the autocorrelation time increases dramatically and can become larger than the total simulation run time. Therefore we have no control over the statistical uncertainties of calculated observable on the simulated lattice configurations. As an example, in lattice QCD as we approach the continuum limit $a \rightarrow 0$ for a fixed physical volume, the computational cost of HMC scales approximately as $a^{-z}$ with $z > 6$ [3]. Several methods in gauge theories has been developed [4–6] to improve the MC simulations on lattice. Machine Learning(ML), in the mean time has made tremendous advancements and found application in many branches in physics. ML has been applied extensively in many condense matter and statistical physics problems [7–10]. In [11], supervised learning has been adopted to accelerate the MC simulations for statistical physics problems, a self learning MC has been proposed in [12] to reduce the autocorrelation time specially near the critical region by learning an effective Hamiltonian. In recent times some machine learning approaches [13–20] are used to circumvent the problem of diverging autocorrelation time in lattice filed theory and XY model [21] as well. Machine Learning(ML) has been also applied to circumvent the problem of critical slowing down in U(1) gauge theory [17] and parameter regression task in [15]. In this work we explore a system with fermions viz. the Gross-Neveu model [22]. In this work, following the ML approach we have used Generative Adversarial Network (GAN) [23] conditioned on parameter of the theory to efficiently generate lattice field configurations near critical point. ML based generative models generates uncorrelated samples which is one of the reason for using it as a replacement of MCMC simulation. To the best of our knowledge, GANs have not been applied to any fermionic system. However, normalizing flows have been used for Yukawa model [14]. In [21], C-GANs have been found to be effective for studying phase transitions in XY model.

The critical point of a lattice theory corresponds to a particular value of the parameters ($\lambda$) of the theory. Our target is to generate uncorrelated samples from a probability distribution

of kind: $P(\Phi|\lambda_{crit}) = \frac{1}{Z}e^{-S(\Phi,\lambda_{crit})}$ where $\Phi$ is lattice field, S is the action and Z is the partition function of the theory. The basic idea of our method consists of the following three steps:

1. Generate samples using HMC for $\lambda$ away from critical region of the lattice theory: $\Phi \sim p(\Phi|\lambda_{noncrit})$.

2. Train the GAN models conditioned on the parameter $\lambda$ using the data from step 1, i.e., learn the distribution $p(\Phi|\lambda)$.

3. Interpolate the trained generator model near the critical point and generate samples from the C-GAN model.

To implement the above ideas we have used a simple field theory in lattice - the Gross-Neveu model(GN model) in 1+1 dimensions [22]. GN model possesses many properties similar to QCD. It is an asymptotically free theory and re-normalizable in 1+1 dimension. GN model undergoes a spontaneous chiral phase transition and is extensively used in the literature as a toy model for QCD. With Wilson fermion, a parity broken phase (Aoki phase) emerges on a finite lattice [24]. The Aoki phase structure of GN model with Wilson fermion and staggered fermion with flavored mass term has been investigated in strong coupling limit in [25]. The chiral phase transition of the GN model with minimally doubled Borici-Creutz fermion has been investigated in detail in [26]. The mass spectrum of GN model has been studied in [27,28].

To check the validity of the generative model, we evaluate it in the critical region. For evaluation, we compare the observables calculated from the samples generated by the proposed C-GAN with those from the samples generated by HMC simulations. Since proposed C-GAN model's samples are independent, given $\lambda$, it alleviates the critical slowing down problem. Since the lattice constant $a$ changes with parameter of the theory, we must choose the lattice size accordingly so that at critical region we get a lattice of desired physical volume.

# 2 Gross-Neveu Model in 1+1 Dimension

## 2.1 Continuum Theory

The Euclidean Lagrangian of GN model in 1+1 dimension is [22]:

$$\mathscr{L} = \sum_{f=1}^{N_f} \overline{\psi}_f(x)(\not{\partial})\psi_f(x) - \frac{g^2}{2}\Big(\sum_{f=1}^{N_f} \overline{\psi}_f(x)\psi_f(x)\Big)^2. \tag{1}$$

With the help of so called Hubbard-Stratonovich(HS) transformation we can reduce the four fermion part to a term quadratic in the fermion fields and an additional auxiliary bosonic field. The transformation is basically a shifted Gaussian integral.

$$\begin{aligned}
&\exp(-\int d^2x[\frac{g^2}{2}(\overline{\psi}_f(x)\psi_f(x))^2] \\
&= \mathcal{N}[\int \mathscr{D}\sigma(x)exp(-\int d^2x[\frac{N_f}{2\tilde{\lambda}}\sigma^2(x) + \overline{\psi}_f(x)\sigma(x)\psi(x)],
\end{aligned} \tag{2}$$

where $\tilde{\lambda} = g^2 N_f$.

The partition function becomes

$$Z = \int \mathscr{D}\overline{\psi}\mathscr{D}\psi\mathscr{D}\sigma e^{-S_\sigma[\overline{\psi},\psi,\sigma]}. \tag{3}$$

The action is given by

$$S_\sigma[\overline{\psi}, \psi, \sigma] = \int d^2x [\frac{N_f}{2\tilde{\lambda}}\sigma^2(x) + \overline{\psi}_f D_{GN}(x)\psi_f(x)], \tag{4}$$

where, $D_{GN} = \slashed{\partial} + \sigma(x)$.

One can show that the $\sigma$ field and condensate field $\overline{\psi}\psi$ are linked via

$$\langle \overline{\psi}(x)\psi(x) \rangle = \frac{-N_f}{\tilde{\lambda}} \langle \sigma(x) \rangle. \tag{5}$$

Therefore, the average of auxiliary field $\langle \sigma(x) \rangle$ can be referred to as the Chiral Condensate. This $\langle \sigma(x) \rangle$ can be used as an order parameter to study spontaneously chiral symmetry breaking of the GN model. GN model is analytically solvable in the infinite flavor limit:$N_f \rightarrow \infty$. Its phase structure has been studied extensively in this limit [29, 30]. Inhomogeneous phases of GN model in lattice are also studied for finite number of flavors with proper continuum limit in [31, 32].

## 2.2 Lattice Theory

The action of lattice GN model in the staggered formalism [33] is generally written as

$$S = \sum_{x,y} [\frac{\lambda N_f}{2}\sigma^2(x) + \sum_{f=1}^{f=N_f} \overline{\chi}_f(x)D(x,y)\chi_f(y)], \tag{6}$$

where the coupling constant is inverted to $\lambda = 1/\tilde{\lambda}$ for simulation purpose and $D = D_1 + \Sigma$ with

$$D_1(x,y) = \frac{1}{2}[\delta_{x,y+\hat{1}} - \delta_{x,y-\hat{1}}] + \frac{1}{2}[\delta_{x,y+\hat{2}} - \delta_{x,y-\hat{2}}], \tag{7}$$

$$\Sigma_{xy} = \frac{1}{4}\delta_{xy}[\sigma(x) + \sigma(x-\hat{1}) + \sigma(x-\hat{2}) + \sigma(x-\hat{1}-\hat{2})], \tag{8}$$

where $\hat{1}$ and $\hat{2}$ are unit vectors in the two directions in 2D.
This theory has discrete chiral symmetry:

$$\chi \rightarrow (-1)^{x_1+x_2}\chi, \quad \overline{\chi} \rightarrow -(-1)^{x_1+x_2}\overline{\chi}, \quad \sigma(x) = -\sigma(x). \tag{9}$$

Higher $N_f$ value is necessary to match continuum ($N_f \longrightarrow \infty$) results but $N_f = 2$ will serve our purpose in this work. After introducing pseudofermionic [34] method(for $N_f = 2$ ) action become non-local:

$$S[\sigma, \phi, \lambda] = \sum_{x,y} [\frac{\lambda}{4}\sigma^2(x) + \phi^\dagger(x)(M^{-1})\phi(y)], \tag{10}$$

where $M = D^\dagger D$ and $\phi$ are pseudofermionic complex field.
Partition function can be written as-

$$Z = \int \mathscr{D}\sigma \mathscr{D}\phi^\dagger \mathscr{D}\phi e^{-S[\phi,\phi^\dagger,\sigma]}. \tag{11}$$

With the action given in Equation (10) we perform our HMC simulations. In this work we have used the staggered fermion(for details about staggered fermion, see [35]) for lattice simulation.

# 3 Generative Adversarial Network (GAN)

Generative Adversarial Networks (GANs) can be trained to generate samples from a high dimensional probability distribution. A GAN [23] basically consist of two neural networks, namely, the generator and the discriminator, where the generator's prime job is to generate realistic samples from a noise vector and discriminator is a binary classifier whose output is either 1 or 0. Notably, GAN learns from the samples from the true distribution, without using the true distribution explicitly. Likewise, it generates samples and does not explicitly tell the probability density.

The Generative model G, parameterized by $\Theta$ is a map from a random noise $z \sim p_z(z)$ to $x \sim p_g(G(z,\Theta))$. The training dataset is from a true distribution $x \sim p_{real}(x)$. The discriminator $D(x,\Phi)$ predict whether it is coming from $p_g$ or $p_{real}$. After completion of the training we expect $p_g$ to be as close as possible to $p_{real}$. The training process is a two players min-max game where discriminator improves its ability to distinguish generator's fake samples and generator also improves its ability to produce more realistic samples to fool the discriminator as the training continues. In the training process both the generator and discriminator's weights are updated in tandem.

The objective function of GAN is:

$$\min_{G} \max_{D} V(G,D) = E_{x \sim p_{real}(x)}[log D(x,\Phi)]$$
$$+ E_{z \sim p_g(z)}[log(1 - D(G(z,\Theta),\Phi))]. \tag{12}$$

**Conditional-GAN:** If the true dataset has categories or classes, then the original GAN approach has no control over the type or class of output generated by the generator as output depends only on the random noise. But in many situations it become necessary to generate data of a particular type or class. So we want to train a GAN so that it can learn a conditional probability distribution.

In C-GAN [36] we append the random noise with additional information $\lambda$, which could be attributes or class labels to produce output $G(\lambda, z, \Theta)$, which is conditioned on $\lambda$. We also append $\lambda$ to the input of discriminator.

The Objective function of C-GAN is:

$$\min_{G} \max_{D} V(\lambda, G, D) = E_{x \sim p_{real}(x)}[log D(\lambda, x, \Phi)]$$
$$+ E_{z \sim p_g(z)}[log(1 - D(\lambda, G(\lambda, z, \Theta), \Phi))]. \tag{13}$$

# 4 HMC Simulation

## 4.1 HMC Algorithm

HMC algorithm can be use to produce a Markov Chain whose stationary distribution is:

$$P(\sigma, \phi | \lambda) = \frac{1}{Z} e^{-S(\sigma, \phi | \lambda)}, \tag{14}$$

where $S(\sigma, \phi)$ is the lattice action and Z is the partition function defined in Equations (10) and (11) respectively. However, this partition function does not represent a classical Hamiltonian system. We can transform it by introducing a canonically conjugate momentum variable $\pi(x)$ into the system. Then it become a Hamiltonian system, where Hamiltonian can be written as:

$$H(\sigma, \pi, \phi) = \frac{1}{2} \sum_{x} \pi^2(x) + S[\sigma, \phi]. \tag{15}$$

In HMC algorithm we solve the Hamiltonian equation in discrete time for $\sigma(x)$ and $\pi(x)$. We can sample pseudofermionic variable $\phi$ easily by sampling complex vector $\xi$ from $exp(-\xi^\dagger \xi)$ and setting $\phi = D^\dagger \xi$. This ensures that $\phi$ can be sampled according to the distribution Equation (14) for a given $\sigma$. For details of HMC for psedofermion action refer to [35]. The common steps one follows in HMC simulation are:

1. Choose $\sigma_0$ configuration from cold-start or hot-start.

2. Choose $\pi$ from random Gaussian distribution.

3. Choose $\xi$ as Gaussian noise and Evaluate:
   $\phi = D^\dagger \xi$.

4. MD steps to update $\sigma$ and $\pi$ keeping $\phi$ as background field: Solve Hamiltonian differential equations for some discrete time step $\tau$.

$$\frac{d}{d\tau}\sigma_x(\tau) = \frac{\partial}{\partial \pi_x}H(\pi(\tau), \sigma(\tau), \phi),$$
$$\frac{d}{d\tau}\pi_x(\tau) = -\frac{\partial}{\partial \sigma_x}H(\pi(\tau), \sigma(\tau), \phi).$$

It will generate new configurations $(\sigma_{new}, \pi_{new})$ as the next proposal.

5. Do Metropolis test to accept or reject the new configuration.

6. Return to step 2.
   In this way we can generate ensemble of $(\sigma, \phi)$ configurations according to the distribution (14).

## 4.2 HMC Simulation and Observables

In this work, we simulate for $N_f = 2$, with lattice size$=32 \times 32$. During HMC simulation, we adjust the MD step-size to keep acceptance rate around $\sim 80\%$ with legitimate autocorrelation time. In this work we set MD step size to 0.1 and trajectory lenght to 1. We left first 500 lattice configurations for thermalization. At each $\lambda$ values in the range [0.6 - 2.5], we generate 4000 lattice configurations for training dataset and baseline for evaluation of our proposed C-GAN, which is discussed further in section VI.

The quantity: $\bar{\sigma} = \frac{1}{N}\sum_x \sigma(x)$, which is measured in a single lattice configuration can be use to study the phase transition as it's ensemble average has direct relation to Chiral Condensate $\langle \bar{\psi}\psi \rangle$. However, there is a problem with quantity $\bar{\sigma}$ as its ensemble average $\langle \bar{\sigma} \rangle$ vanishes even for $\lambda \lessgtr \lambda_{crit}$ i.e. even for broken phase close to the critical point. This can be seen from Figure 1 which is calculated near critical point where configurations fluctuates between two minimum and hence average $\langle \bar{\sigma} \rangle$ nearly vanishes. This is due to the ability of configurations to make tunnel from one minimum to the other. So instead of using $\langle \bar{\sigma} \rangle$, we choose $\langle |\bar{\sigma}| \rangle$ as our order parameter which is a suitable observable to study phase transition. One more observable of importance is susceptibility. The two observables can be defined as

$$\langle |\bar{\sigma}| \rangle = \frac{1}{N}\langle |\sum_x \sigma(x)| \rangle, \quad \chi = N[\langle \bar{\sigma}^2 \rangle - \langle |\bar{\sigma}| \rangle^2], \tag{16}$$

where $N$ is the lattice volume.

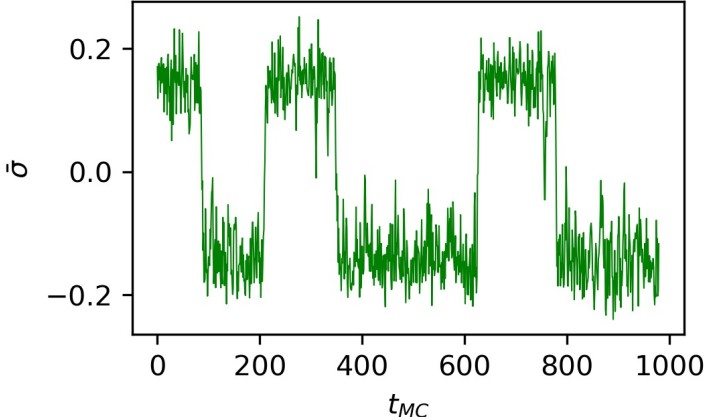

Figure 1: Fluctuations of $\bar{\sigma}$ values for lattice configurations during HMC simulation in between two minima at $\lambda \lessgtr \lambda_{crit}$.

## 5 Proposed Method

In HMC simulation we sample $\sigma, \phi$ field according to the the distribution given in Equation (14) i.e. $\sigma, \phi \sim P(\sigma, \phi | \lambda)$. We want the C-GAN to learn the marginal distribution of $\sigma$ field i.e $p(\sigma | \lambda)$. So we discard HMC generated pseudofermionic $\phi$ samples and only consider $\sigma$ samples which now represent the marginal distribution of $\sigma$ from the joint distribution in Equation (14). Let the samples from the C-GAN represent an implicit distribution $\hat{p}(\sigma | \lambda)$. Our target is to train the C-GAN so that $\hat{p}(\sigma | \lambda)$ approximates the true distribution $p(\sigma | \lambda)$. To address the problem of critical slowing down, we train the C-GAN model for $\lambda$ values sampled in non-critical region, where the autocorrelation time is much smaller comparing to the critical region. Hence generation of training dataset is not affected by critical slowing down. Then we use the trained C-GAN model to generate samples near critical $\lambda$. Since C-GAN model generates independent samples, hence our method can produce uncorrelated lattice configurations in the critical region.

Vanilla C-GAN trained over the HMC samples fails to learn the distribution reliably. The learning is made efficient as well as robust by incorporating into the C-GAN model the information of symmetries and constraints in the theory. Also, transforming the samples so as to reduce the imbalance in their values improves learning. We discuss these in detail in the following subsections.

### 5.1 Translation Symmetry

Due to translation symmetry in GN model lattices, C-GAN generator made of dense layers fails to learn the true distribution properly. Convolutional kernels allow translational invariance in the lattices. Hence, using convolutional layers in the generator allows the learning to take place efficiently.

### 5.2 Transformation of $\sigma$ Field

Since the observables of GN model can be calculated from $|\bar{\sigma}|$, hence for training the C-GAN we transformed the lattice configurations such that each configuration has $\bar{\sigma} > 0$. This will reduce degrees of freedom for the C-GAN model which will help in exploring the distribution space more efficiently. For that purpose, we select a particular configuration and if found $\bar{\sigma} < 0$

then we apply a transformation:

$$\sigma(x) = -\sigma(x), \quad \forall x. \tag{17}$$

For training purpose of C-GAN, we apply natural log transformation to the HMC generated samples as follows:

$$\sigma_i'(x) = ln(\sigma_i(x) + c), \quad \forall x,i, \tag{18}$$

where i represent a single lattice configuration from the ensemble and c is a constant such that the sum inside the logarithm become positive. This transformationEquation (18) become necessary for stable training of C-GAN as it balances data values and reduces the dynamical range of the $\sigma(x)$ field.

For efficient training we apply the Min-Max scaling to the above transformed data to bring into a range [-1,1].

## 5.3 Periodic Boundary Condition

During generation of configurations by HMC, we apply periodic boundary condition i.e. we replace $\sigma(i,j)$ by $\sigma'(i,j) := \sigma((i)_N, (j)_N)$, where $(i)_N$ represents $i$ modulo $N$. In order to learn the periodicity by the C-GAN model we apply periodic padding to the all layers of generator and initial two layers of discriminator.

# 6 Numerical Experiment & Results

## 6.1 Dataset

Our training dataset is consisting of 10 ensembles, each of which has 4000 lattice configurations corresponding to 10 different $\lambda$ values generated by HMC simulation. $\Lambda_{tr}$ is the set of $\lambda$ on which we train the C-GAN model. It includes $\lambda$ values away from the critical region. Assuming $\lambda_{crit} \sim 1.5$, $\Lambda_{tr} = \{0.6, 0.8, 1.0, 1.2, 1.3, 1.8, 2.0, 2.2, 2.3, 2.5\}$. For inference and evaluation of the proposed C-GAN model we use $\Lambda_{ts}$ which includes $\lambda$ values in the critical region too, $\Lambda_{ts} = \{0.6, 0.8, 1.0, 1.3, 1.5, 1.6, 1.7, 2.0, 2.3, 2.5\}$.

## 6.2 C-GAN Model Architecture

Input to the generator model consist of a $4 \times 4$ i.i.d Gaussian noise with zero mean and unit variance, stacked together with $4 \times 4$ matrix containing all entries as $\lambda$. In generator model three 2D Transposed convolutional layers are used for up-sampling to $32 \times 32$ final lattice configuration. We use kernel of sizes (3,3) & (4,4) and strides (1,1) & (2,2) for generator model. In discriminator the input is a grid of $32 \times 32$ $\sigma$ samples, concatenated with a $32 \times 32$ channel with the $\lambda$ value repeated in all the cells. It has three 2D convolutional layers with Tanh activation function followed by a dense layer with Sigmoid activation. We use kernel of sizes (4,4) and strides (2,2),(1,1) for discriminator.The detail architecture of generator and discriminator model is given in the appendix. We add periodic padding to the all layers of generator model and only two initial layers of discriminator model to learn the periodicity in the lattice configuration.

Once the training of C-GAN is over, we use the generator model to generate two ensembles each consist of 20000 configurations for $\Lambda_{tr}$ and $\Lambda_{ts}$ respectively. In both cases we evaluate our C-GAN model by comparing observables calculated on the above two set with those calculated from the HMC generated ensembles. The observables used for this purpose are: $\langle|\bar{\sigma}|\rangle$ and $\chi$ as defined in Equation (16).

### 6.3 C-GAN Training and Sampling Process

In the preperation of datasetfor training, we put $\lambda$ label for each HMC generated configurations which are the real data in discriminator terminolgy.

While updating the generator one batch of random noise $z$ (with random label) is drawn from $p_g(z)$. For updating discrimintor, a full batch of 256 configurations is used, where half batch from $x = G(z|\lambda)$, where $z \sim p_g(z)$ and other half from $x \sim p_{real}(x|\lambda)$. In this work we use Adam optimizer with an initial learning rate of 0.0002 for both geneartor and discriminator loss. Initially the C-GAN model was trained upto 200 epochs. Then we use observables errors, $\delta\sigma = \langle|\bar{\sigma}|\rangle_{hmc}$-$\langle|\bar{\sigma}|\rangle_{C-GAN}$ as stoping criteria. We stopped the trainig where the error on validation set is minimum and remains approximatly constant for 10 further epochs. However, this epoch range will change as the learning rate, optimizer, batch size, number of weights and biases, padding structure etc. in the C-GAN model changes.The loss curve is shown in Figure 2. For sampling purpose, we choose a particular $\lambda$ value of shape ($4 \times 4$) and a batch

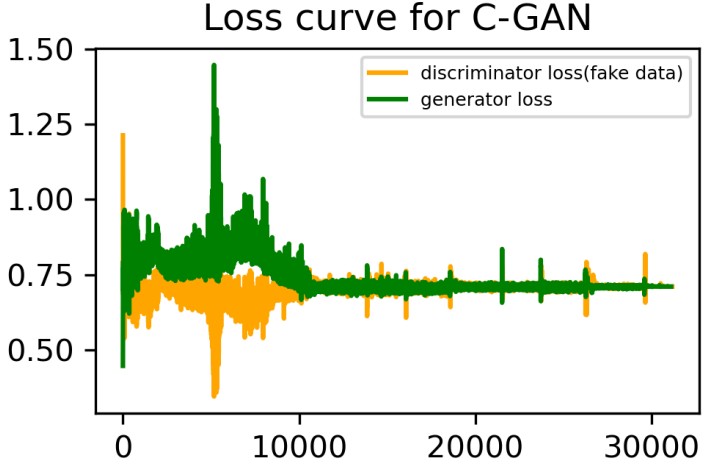

Figure 2: **C-GAN loss curve.**

from $z \sim p_g(z)$ which is then fed to the pretrained generator model. The batch size is the number of samples required to generate which is 2000 for our case. We generate total 20000 configurations for different $\lambda$ values for both phase case. For interpolation and etxrapolation at critical $\lambda$ values we use the same sampling procedure.

### 6.4 Results

#### 6.4.1 Testing on $\Lambda_{tr}$ Set

We do the analysis on $\lambda_{tr}$ set to confirm that the C-GAN model has correctly learned the training data distribution. In this ensemble we calculate the $\bar{\sigma}$ for each lattice configuration then plot the histogram of $|\bar{\sigma}|$ as shown in Figure 3. Different peaks in the histogram roughly corresponds to different $\lambda$ values. The histogram generated from the proposed C-GAN model and HMC overlaps quite well. It indicates that our proposed distribution $\hat{p}(\sigma|\lambda)$ represented by C-GAN approximates the true distribution for the $\lambda_{tr}$ set. Also in Figure 4, we plot ensemble averaged $\langle|\bar{\sigma}|\rangle$ for $\Lambda_{tr}$ set. Here we take ensemble average $\langle|\bar{\sigma}|\rangle$ for each $\lambda$ separately and then plot $\langle|\bar{\sigma}|\rangle$ vs $\lambda$. It shows that the observables are matching well for both C-GAN and HMC ensembles for $\Lambda_{tr}$ i.e. the set used during training.

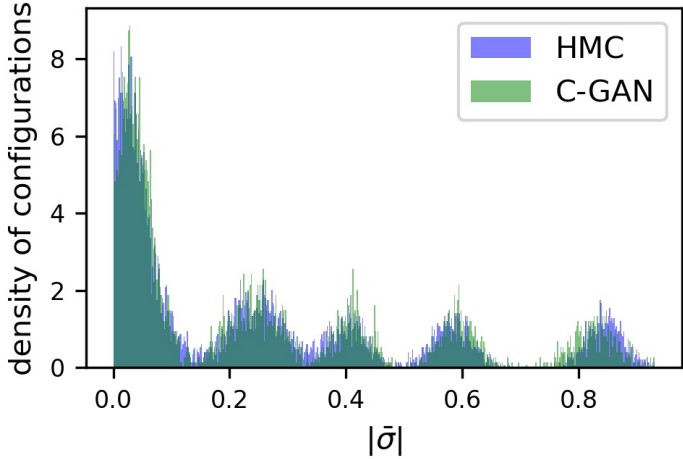

Figure 3: Histogram of $|\bar{\sigma}|$ for $\lambda_{tr}$ set, estimated from samples obtained via HMC and C-GAN.

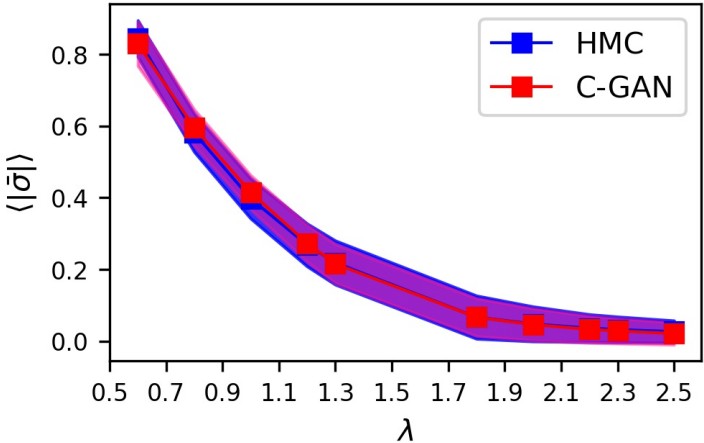

Figure 4: Mean $\langle|\bar{\sigma}|\rangle$ and standard deviation on $\lambda_{tr}$ set, estimated from samples obtained via HMC and C-GAN.

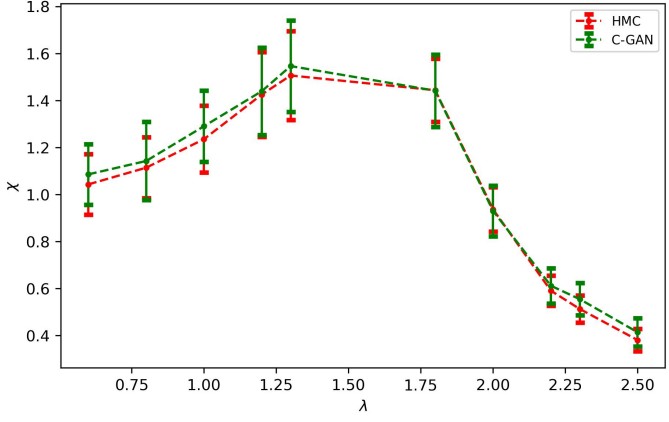

Figure 5: Susceptibility and its standard deviation on $\lambda_{tr}$ set, estimated from 8000 samples with bin size of 100.

### 6.4.2    Testing on $\Lambda_{ts}$ Set

Since our main goal is to generate lattice ensembles in the critical region we must evaluate our C-GAN model in $\Lambda_{ts}$. We asses the performance of the proposed C-GAN in terms of being able to produce observables matching with those obtained from true distributions(i.e., generated by HMC).

**Mean**$\langle|\bar{\sigma}|\rangle$**:** In Figure 6 we can observe that histogram of $|\bar{\sigma}|$ matches quite well with the true histogram obtained via HMC samples even for $\Lambda_{ts}$. Different peaks at high $|\bar{\sigma}|$ values roughly represent different $\lambda$ values. However, there are no distinct peaks visible near low $|\bar{\sigma}|$ values as the peaks gets overlapped. We also present the histograms of $|\bar{\sigma}|$ in Figures 7 and 8 for $\lambda \in \{1.5, 1.6\}$ which are in the critical region where we didn't train the model. In Figure 9 we present the results for the mean $\langle\bar{\sigma}\rangle$ in the critical region. We can see that the phase transition behaviour is described very well by the generator model.

**Susceptibility($\chi$):** We show the susceptibility values obtained from HMC configurations as well as those obtained from C-GAN in Figure 5 for non-critical data set. One can observe that the peak coincides for both HMC and C-GAN. The same plot for critical dataset is shown in Figure 10. We have found that in critical region both mean $\langle\bar{\sigma}\rangle$ and susceptibility agree quite well with the HMC results even without training in that region. This gives a good indication that the trained model can reproduce the second order phase transition in the GN lattice model.

In Figure 11 we show the autocorrelation time generated from HMC simulation with unit trajectory length in MD step, while keeping acceptance rate $\approx 80\%$. We see that near the phase transition point the the autocorrelation time increases sharply. However, during sampling from the C-GAN model we starts with a random Gaussian noise vector to generate lattice configurations. Therefore, the lattice configurations generated by the C-GAN model are independent of each other, which will solve the critical slowing down problem. In this way we can generate uncorrelated samples near critical region at the cost of generation of samples by HMC at the non critical region.

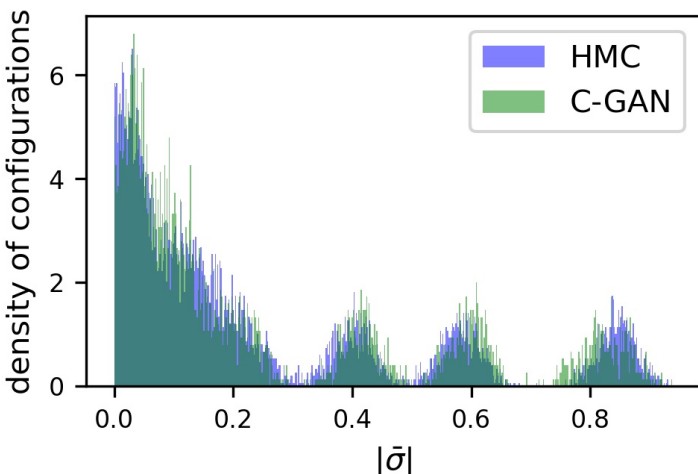

Figure 6: Histogram of $|\bar{\sigma}|$ for$\Lambda_{ts}$ set, estimated from samples obtained via HMC and C-GAN.

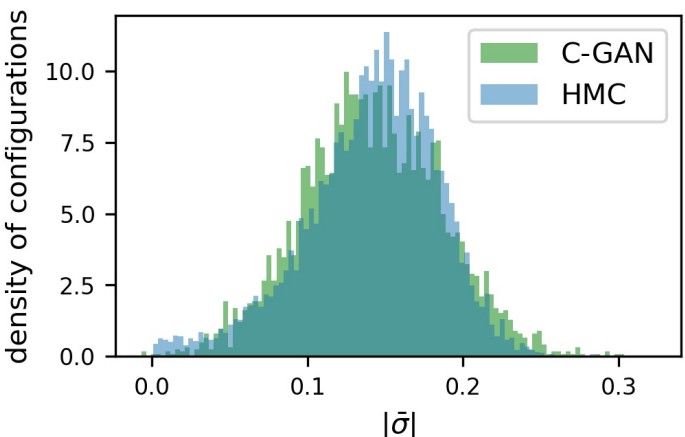

Figure 7: Histogram of $|\bar{\sigma}|$ at $\lambda = 1.5 \in \Lambda_{ts}$: HMC and C-GAN histograms overlaps quite well.

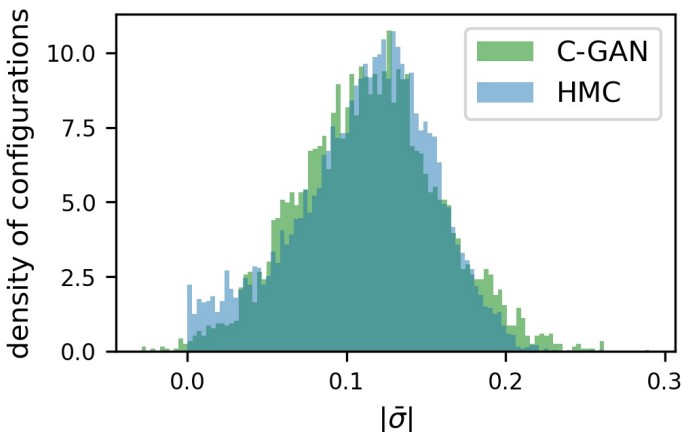

Figure 8: Histogram of $|\bar{\sigma}|$ at $\lambda = 1.6 \in \Lambda_{ts}$: HMC and C-GAN histograms overlaps quite well.

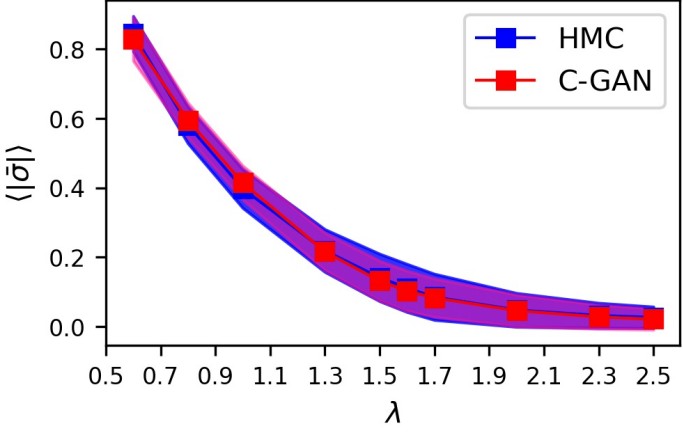

Figure 9: Mean $\langle|\bar{\sigma}|\rangle$ and standard deviation for $\Lambda_{ts}$ set, estimated from samples obtained via HMC and C-GAN.

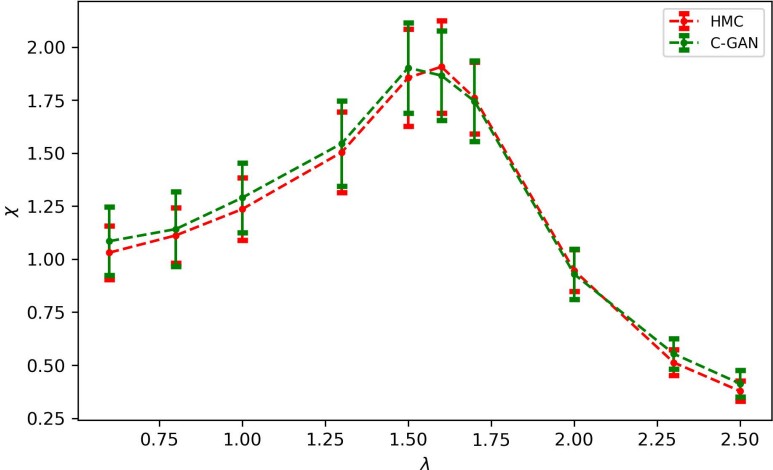

Figure 10: Susceptibility and its standard deviation on $\lambda_{ts}$ set, estimated from 8000 samples with bin size of 100.

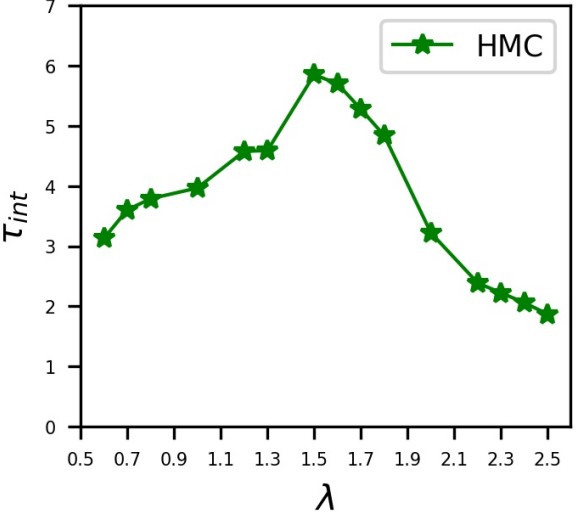

Figure 11: Integrated Autocorrelation time estimated from HMC simulation with MD trajectory length, $\tau = 1$.

## 6.5 Numerical Experiment with Data from a Single Phase

We also train the C-GAN model using HMC generated dataset consiting of $\lambda$ values only from one single phase. This experiment is necessary to check our model's utility in latttice gauge theory where extrapolation to critical point is necessary from one direction of parameter space.

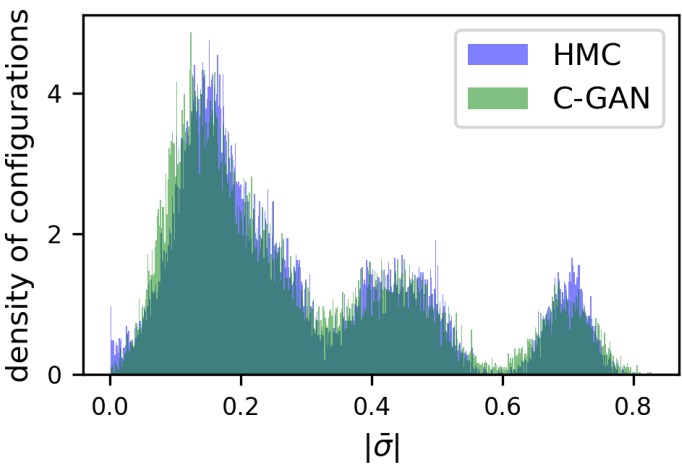

Figure 12: Histogram of $|\bar{\sigma}|$ for $\Lambda^{tr}_{1ph}$ set, estimated from samples obtained via HMC and C-GAN.

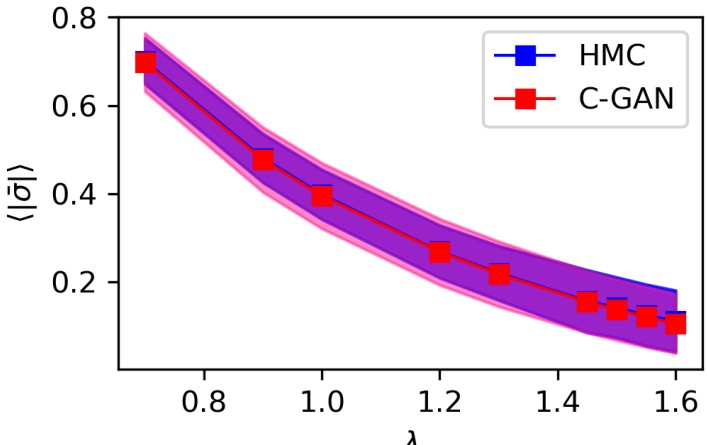

Figure 13: Mean $\langle|\bar{\sigma}|\rangle$ and standard deviation for $\Lambda^{ts}_{1ph}$ set, estimated from samples obtained via HMC and C-GAN.

The training set of $\lambda$ values are $\lambda^{tr}_{1ph} = \{0.4, 0.6, 0.7, 0.8, 0.9, 1.0, 1.1, 1.2, 1.25, 1.3, 1.35, 1.4\}$ taken from the broken phase and the test set of $\lambda$ values are $\lambda^{ts}_{1ph} = \{0.7, 0.9, 1.0, 1.2, 1.3, 1.45, 1.5, 1.55, 1.6\}$. We extrapolate the C-GAN model to critical region of $\lambda$ values 1.45, 1.5, 1.55 and 1.6.

The results are shown in Figures 12 and 16. We observe that the histogram and mean $\langle\bar{\sigma}\rangle$ matches quite well with HMC results for $\lambda^{ts}_{1ph}$, where critical points are included. Also in Figures 13 to 15 we have shown the individual histogram of $|\bar{\sigma}|$ for $\lambda = 1.5, 1.55, 1.6$. We found that for the critical $\lambda$ values observables does't differ either we train the model with one single phase or consider both the phases.

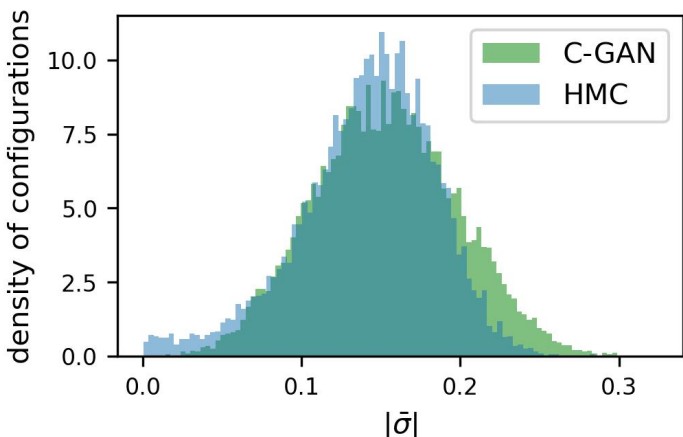

Figure 14: Histogram of $|\bar{\sigma}|$ at $\lambda = 1.5 \in \Lambda_{1ph}^{ts}$: HMC and C-GAN histograms overlaps quite well.

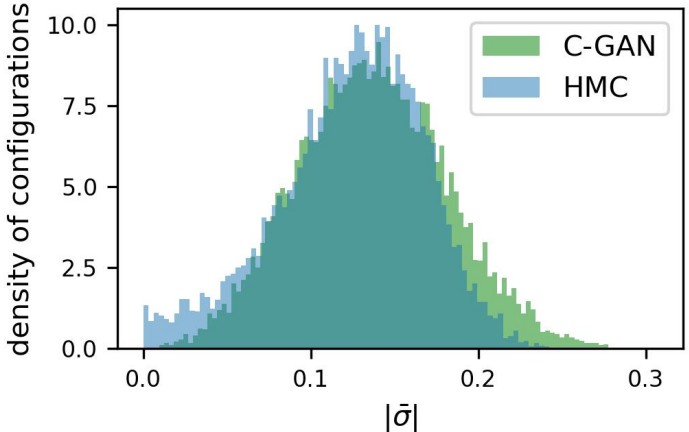

Figure 15: Histogram of $|\bar{\sigma}|$ at $\lambda = 1.55 \in \Lambda_{1ph}^{ts}$: HMC and C-GAN histograms overlaps quite well.

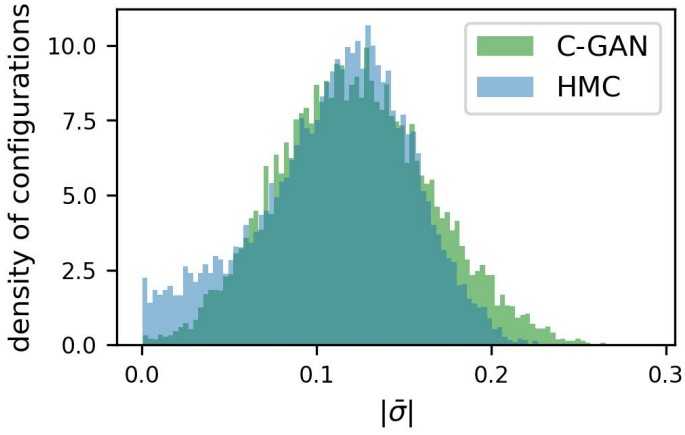

Figure 16: Histogram of $|\bar{\sigma}|$ at $\lambda = 1.6 \in \Lambda_{1ph}^{ts}$: HMC and C-GAN histograms overlaps quite well.

## 6.6   Ablation Analysis

We perform ablation analysis to see the effect of certain key component of the proposed method on its performance.

**Transformation of $\sigma$ field:** We find that log transformation Equation (18) is one of the crucial component for the training of C-GAN model. On removing it, the training loss becomes high and the observables do not agree well with the HMC observables which can be seen from Figures 17 to 19. There is a large deviation of mean of $|\bar{\sigma}|$ compared to HMC results in both critical and non-critical regions as shown in Figure 17. It is observed that susceptibility is too sensitive to log transformation as shown in Figure 18. It is also seen that without Min-Max scaling the C-GAN model is unable to learn different modes corresponding to different $\lambda$ values.

**Periodic Boundary Condition:** We have noticed that the C-GAN performs well using periodic padding in both discriminator and generator as seen from Figures 6 to 10. But when we remove periodicity from both discriminator and generator, C-GAN fails to reproduce the HMC results. Figures 20 and 21 show the disagreements between C-GAN and HMC results for $\langle|\bar{\sigma}|\rangle$, susceptibility $\chi$ respectively and CrefFig16 compares the histogram of $|\bar{\sigma}|$ without periodicity in C-GAN for $\lambda = 1.5$. Likewise, not applying periodic padding to the generator and applying only to the discriminator, also degrades the performance.

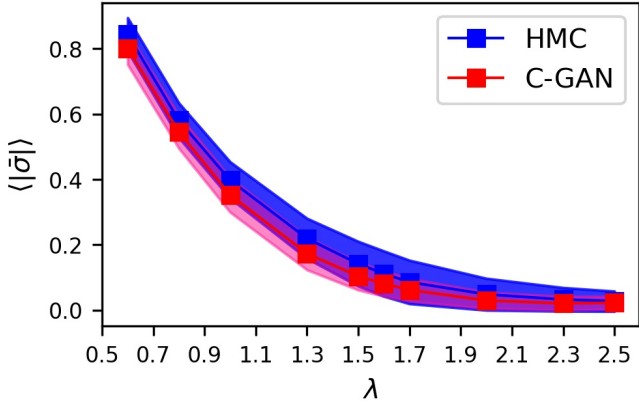

Figure 17: Ablation for log transformation: Mean $\langle|\bar{\sigma}|\rangle$ and standard deviation on $\Lambda_{ts}$ set without log transformation.

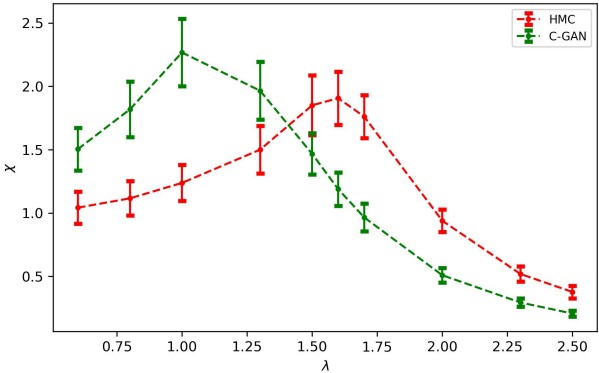

Figure 18: Ablation for log transformation: Susceptibility on $\Lambda_{ts}$ set without log transformation.

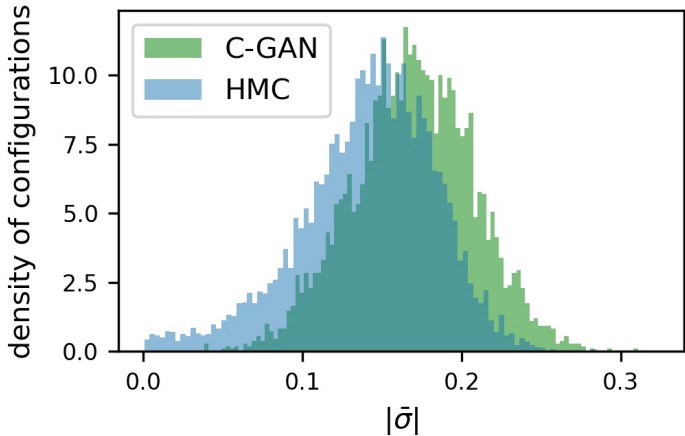

Figure 19: Ablation for log transformation: Histogram of $|\bar{\sigma}|$ at $\lambda = 1.5 \in \Lambda_{ts}$ set without log transformation.

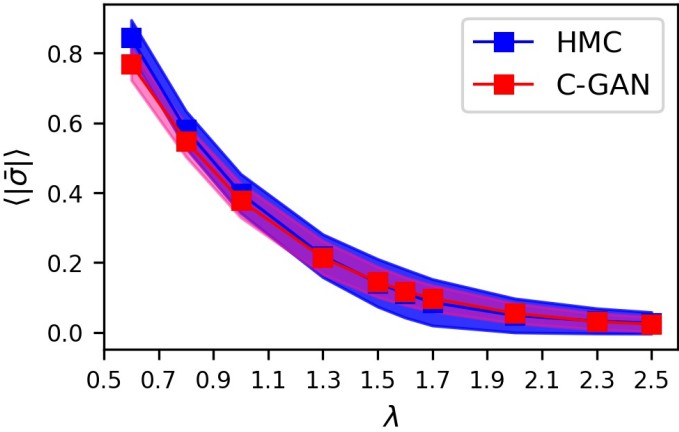

Figure 20: Ablation for periodic padding: Mean $\langle|\bar{\sigma}|\rangle$ and standard deviation on $\Lambda_{ts}$ set without periodic padding in both generator and discriminator.

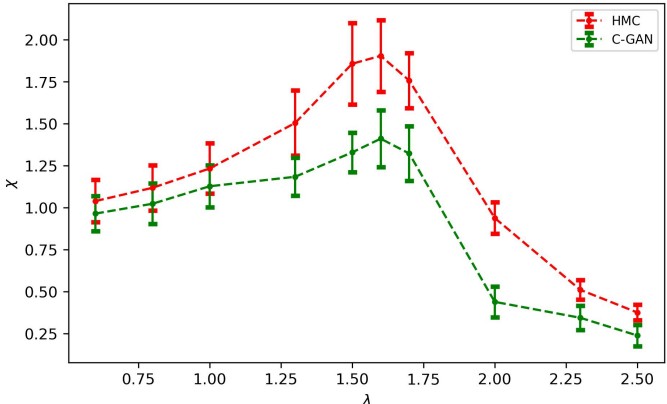

Figure 21: Ablation for periodic padding: Susceptibility on $\Lambda_{ts}$ set without periodic padding in both generator and discriminator.

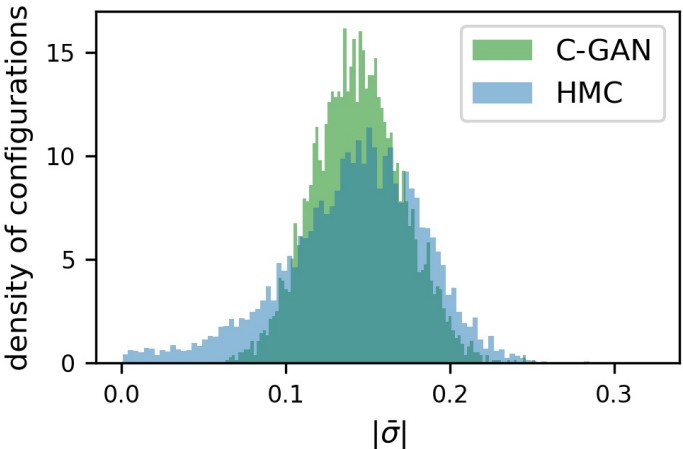

Figure 22: Ablation for periodic padding: Histogram of $|\bar{\sigma}|$ at $\lambda = 1.5 \in \Lambda_{ts}$ set without periodic padding in both generator and discriminator.

## 6.7 Cost Analysis

Sampling a fixed numbers of configurations via HMC algorithm depends on various parameters like MD step size, no. of MD steps, parameter value of action and also hardware used for simulations. Here we have used MD stepsize=0.1 and no. of MD steps=10. The hardware used for HMC simulations is a six core i7-9700CPU CPU machine. In non-critical region we generate around 10000 configurations per hour, leaving 10 intermidiate configurations. In critical region, we leave 20 intermediate configurations and able to generate roughly 4000 configuration per hour.

The training of C-GAN model was done on a single GPU machine (GeForce RTX) for 2-3 hours[1]. Once the training is over, sampling of lattice configurations become very efficient. It roughly takes only 2 minutes to generates 8000 lattice configurations.

These gains looks significant and we expect them to be more significant as we go to the higher dimension where autocorrelation for HMC simulation is more severe near critical region.

## 7 Summary & Conclusion

MCMC methods are generally used to generate lattices as they give theoretical guarantees on validity of samples. In this work, we use GANs which don't give theoretical guarantees but the empirical results show that they are able to efficiently interpolate as well as extrapolate to critical regions. In lattice field theory, the cost of generation of lattice configurations by MCMC methods is severely affected by critical slowing down as the lattice parameters are tuned towards the critical region. At the critical point the cost of HMC simulation diverges for theories like QCD due to the diverging autocorrelation time. Therefore, generation of configurations in lattice field theory in the critical region is a challenging task. This paper proposes to use HMC generated configurations for GN model away from the critical region and trained a C-GAN to generate lattice configuration near critical point. With HMC data in non-critical region, we train the C-GAN model conditioned with parameter $\lambda$. For evaluation of the proposed C-GAN model at critical region we compare few observables on the samples generated from both C-GAN and HMC. We found a good matching between the results of HMC and our C-GAN model and also observed that phase transition can be very well reproduced by the generative

---

[1]We have used tensorflow 2.4 for our model implementation.

approach. Since the C-GAN model in the critical region gives correct critical behaviour, we can infer that our generative model is a good interpolator in the critical region. Since C-GAN generates independent configurations, there is no correlation in the samples generated by the C-GAN model, thereby avoiding the critical slowing down problem. In this work we evaluate our proposed C-GAN model by comparing observables with HMC samples. We could also use our C-GAN model distribution $\hat{p}(\sigma|\lambda)$ as proposal distribution to construct a Markov chain as done in [13]. However, to construct such a Markov chain we must know the proposal density explicitly which is not available for GANs. Rather in this work we have accepted all the samples generated by the C-GAN model and thus having a vanishing autocorrelation. However, we can construct a markov chain in future looking at the recent development regarding density estimation for GAN in ML community. One such method could be the FlowGAN [37] which explicitly estimate the densities for GAN, where generator network is replaced by an invertible flow network. Another method could be the Round-trip method [38] which try to estimate density approximately. There are also other ML architecture like Conditional Normalizing Flow which estimate densities explicitly for generated samples. These are few possibility in ML architecture which can be use for MCMC accept/reject step and we are planing to work in these directions.

Although the problem of critical slowing down is not as severe for GN model in 1+1 dimensions but building and testing the C-GAN in the GN model establishes its applicability in the lattice formulation of fermionic system. In this work, we dealt with only fermionic fields without any gauge interaction. Extending our work to lattice gauge theory and QCD will be an interesting as well as challenging task.

## Acknowledgments

Thanks to Prof. William Detmold for helpful discussions, and to SPARC for facilitating his visit at IIT Kanpur.

## Appendix

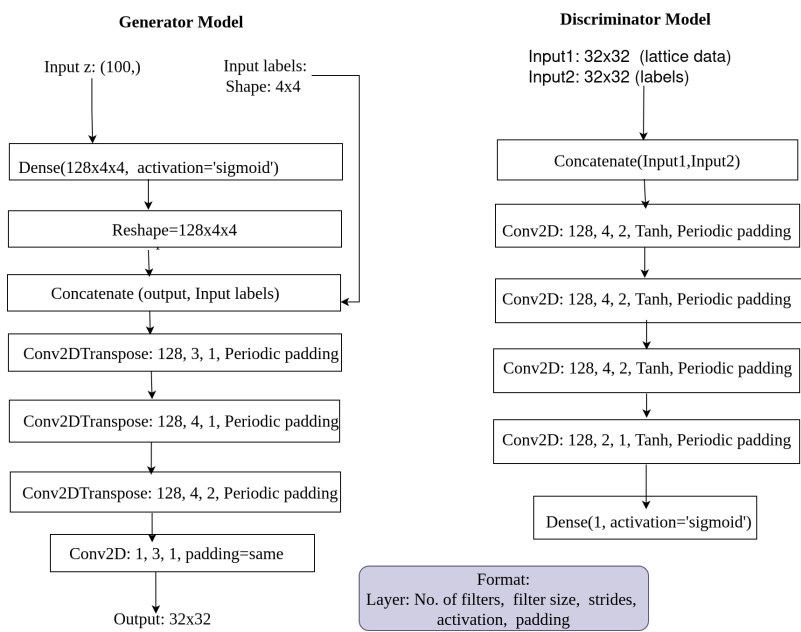

Figure 23: **Architecture of C-GAN models: Generator and Discriminator model.**

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
