# Peer review of "Generative learning for the problem of critical slowing down in lattice Gross Neveu model"

_SciPost Physics, doi:SciPost Phys. Core 5, 052 (2022)_

## Round 1 · Referee Report · Anonymous (Referee 1) · 2022-3-22

Report

The paper concerns the application of a C-GAN to the sampling of lattice field configurations in the (1+1)d Gross-Neveu model near criticality. The investigation of generative neural samplers for the mitigation of critical slowing down is timely and the authors successfully improve upon previous works applying GANs to lattice field theory. The present work is also the first instance of this architecture being applied to a fermionic system. The presentation of the results is clear and concise.

It is impressive that the model manages to largely reproduce important observables close to the critical point without explicitly being trained there, because the relevant information is strongly suppressed away from the critical region. The observation that the C-GAN can extract this information is non-trivial and emphasizes the potential of conditional architectures for the extrapolation to different action parameters. The discussion about the necessity to implement the symmetries of the studied theory into the network in order to achieve these results is very interesting.

Nevertheless, one of the C-GAN's obvious shortcomings is its inability to provide explicit probabilities for the generated configurations. This makes guaranteed asymptotically exact sampling difficult due to the missing notion of importance. Hence, it is basically impossible to decide whether extrapolated results are in fact trustworthy without also performing standard MCMC calculations in the region of interest, which defeats the original purpose of training the C-GAN. While the proposed approach generates statistically independent samples and is therefore not affected by critical slowing down, this comes at the cost of sacrificing exactness. The authors already acknowledge this in the conclusion; however, some further comments regarding potential solutions to this problem may be appropriate. For example, there have been several works in the ML community on likelihood estimation for GANs, which could be cited here. It might also be useful to mention other generative ML architectures that can provide tractable probability densities as potential targets for future work.

Further, it would be very helpful to have some kind of computational cost comparison between the proposed sampling approach and HMC. While precise statements are certainly difficult due to the algorithms being very different, rough estimates of the training/sampling/simulation times would already have merit. It might also be useful to state what software and hardware was used in the process.

In order to increase reproducibility, the authors could also provide more details about the training/sampling procedures and hyperparameter settings, including: type of optimizer, learning rate, number of epochs, batch size, loss curves, number of MD steps per trajectory and step size, etc.

In Section III, it could be mentioned more clearly that the discriminator is a binary classifier and therefore its output only takes the values 0 or 1. This would help readers unfamiliar with GANs to better understand the objective functions.

Finally, there are a few minor issues with punctuation and whitespace. In particular, whitespace before citations as well as parentheses is inconsistent, ideally there should be a single space everywhere. This also applies to references like Eq. () or Fig. (), and in some cases the parentheses are missing. Here it might be useful to just employ the cleverref package with the command \Cref{}. There should be no space before colons (:). The abbreviation "Ref."/"Refs." is used in some cases, but not others. Papers cited together should ideally be in a single \cite{} environment, e.g. in the first paragraph it should be [4-6] instead of [4][5][6]. A period ending the sentence is missing right below Eq. (4), as well as in the captions of most figures. In Section II, there is a typo in the word "auxialary", which should be "auxiliary". Colons in section and subsection titles seem unnecessary, e.g. in the title of the conclusion section. Fixing these issues would improve the readability of the manuscript.

In summary, while the paper is certainly of high quality, I do not think it meets the standards of SciPost Physics. However, I would definitely recommend this paper for publication in SciPost Physics Core if the above points are addressed.
  • validity: -
  • significance: -
  • originality: -
  • clarity: -
  • formatting: -
  • grammar: -

Author:  Ankur Singha  on 2022-04-04  [id 2352]

(in reply to Report 1 on 2022-03-22)

We are thankful to the Referee for the remarks made on the manuscript. Accordingly, we have modified the manuscript and mention the changes we have made.
These are our responses:

We agree with the referee that one of the obvious shortcomings of C-GAN is its inability to provide explicit probabilities for the generated configurations, for which we are unable to perform the metropolis accept/reject test. But there have been some recent works that attempt to estimate the likelihood for GANs and this can form the basis of our investigation. So, following the advice of the referee, we have added some additional text in section VII to better explain the potential solution for this problem and added 3 references related to the density estimation for GANs.

As suggested by the referee, we have added one subsection for cost analysis, in Section VI. In this section, we have mentioned the time required for data generation in HMC simulation, training duration for C-GAN model and data generation time from the pre-trained model. We also mentioned the software/hardware used for both kinds of simulation.

We have also added one subsection to explain the training and sampling process in section VI to increase the reproducibility of our paper results, where we have mentioned the various hyperparameters used in this work and also added the training loss curve in section VI.

We have included the line "The discriminator is a binary classifier and therefore its output only takes the values 0 or 1" in section III, as suggested by the referee.

We are grateful that the referee points out these errors. We have gone through the paper and fixed punctuation and white space errors, and also used the cleverref package to refer to the equations and figures. Furthermore, we have used only a single cite environment for citation and corrected the typo mentioned.

Attachment:

---

## Editorial Decision

published